# Deep Learning for Computing Convergence Rates of Markov Chains

**Yanlin Qu**     **Jose Blanchet**     **Peter Glynn**
Department of Management Science and Engineering
Stanford University
{quyanlin,jose.blanchet,glynn}@stanford.edu

## Abstract

Convergence rate analysis for general state-space Markov chains is fundamentally important in operations research (stochastic systems) and machine learning (stochastic optimization). This problem, however, is notoriously difficult because traditional analytical methods often do not generate practically useful convergence bounds for realistic Markov chains. We propose the Deep Contractive Drift Calculator (DCDC), the first general-purpose sample-based algorithm for bounding the convergence of Markov chains to stationarity in Wasserstein distance. The DCDC has two components. First, inspired by the new convergence analysis framework in (Qu et al., 2023), we introduce the Contractive Drift Equation (CDE), the solution of which leads to an explicit convergence bound. Second, we develop an efficient neural-network-based CDE solver. Equipped with these two components, DCDC solves the CDE and converts the solution into a convergence bound. We analyze the sample complexity of the algorithm and further demonstrate the effectiveness of the DCDC by generating convergence bounds for realistic Markov chains arising from stochastic processing networks as well as constant step-size stochastic optimization.

## 1   Introduction

General state-space Markov chains are indispensable in a wide array of fields due to their flexibility and applicability in modeling random dynamical systems. To analyze the long-term behavior of these Markovian models, estimating the rate of convergence to equilibrium is critical. When designing reliable real-world systems (e.g. cloud platforms and manufacturing lines), the faster the convergence, the faster the recovery after disturbances. When designing efficient sample-based algorithms (e.g. stochastic gradient descent (SGD) variants and MCMC), the faster the convergence, the faster the goal attainment. The rate of convergence also appears in MDP-related sample complexity results under the name "mixing time". Although convergence rate estimation is critically important, estimating the convergence rate of even a mildly complex chain can be extremely difficult.

Over the last three decades, significant efforts have been made to bound the convergence of general state-space Markov chains. Most of these works utilize a pair of drift and minorization conditions (D&M) to bound the convergence in terms of the total variation (TV) distance (Meyn et al., 1994; Rosenthal, 1995; Jarner and Roberts, 2002; Douc et al., 2004; Baxendale, 2005; Andrieu et al., 2015). The drift condition forces the chain to move towards a selected region. On such a region, the minorization condition allows the chain to regenerate or to couple with a stationary version of the chain. This analysis tends to produce overly conservative TV bounds, especially in high-dimensional settings; see (Qin and Hobert, 2021) for a discussion.

The Wasserstein distance, as a measure of convergence to equilibrium, can exhibit better dimension dependence (Qin and Hobert, 2022b). In addition, many Markov chains of interest (e.g. constant

step-size SGD minimizing convex loss on finite datasets) converge in Wasserstein distance but not in TV distance. Consequently, bounding convergence in Wasserstein distance has steadily gained popularity over the years (Gibbs, 2004; Hairer et al., 2011; Butkovsky, 2014; Durmus and Moulines, 2015; Durmus et al., 2016; Qin and Hobert, 2022a). Most of these works replace the minorization condition with a contraction condition (D&M becomes D&C). After returning to a selected region, two copies of the chain tend to become closer to each other. Both D&M and D&C enforce two conditions in two respective regions. However, partitioning the state space into two distinct regions often leads to suboptimal rates.

Recently, (Qu et al., 2023) introduce the so-called contractive drift condition (CD), a single condition enforced on the entire state space, to explicitly bound the convergence in Wasserstein distance. A special case of CD dates back to (Steinsaltz, 1999). By verifying CD, (Qu et al., 2023) establish parametrically **sharp** convergence bounds for stylized Markov chains arising from queueing theory and stochastic optimization (e.g. revealing how step-size, heavy-tailed gradient noise, growth rate and local curvature of objectives affect the convergence of stylized SGD). Although CD may generate better bounds than D&M and D&C for stylized chains (e.g. SGD with iid gradient noise), these methods are generally intended as theoretical tools that can provide closed-form convergence bounds for structured models. For more realistic, less structured chains, computational rather than analytical methods are needed. However, despite of the rapid development of computational power in the past decade, the convergence analysis of general state-space Markov chains is still in the pen-and-paper age.

To launch **computational** Markov chain convergence analysis, we need a key to switch on the deep learning engine. This paper introduces the *Deep Contractive Drift Calculator* (DCDC) is the first general-purpose sample-based algorithm for bounding the convergence of general state-space Markov chains. There are two key ideas we develop. The first is to observe that CD, an inequality by definition, is actually an equality by nature (if the inequality has a solution, then the corresponding equality also has a solution). Thus, we introduce the Contractive Drift Equation (CDE), an integral equation the solution of which leads to an explicit convergence bound. For the second part, inspired by the success of physics-informed neural networks (PINNs) in solving PDEs (Sirignano and Spiliopoulos, 2018; Raissi et al., 2019), we develop an efficient neural-network-based CDE solver. By combining these two components, DCDC solves CDEs by training neural networks and converts solutions into explicit convergence bounds. DCDC demonstrates the potential of computer-assisted convergence analysis and bridges the gap between deep learning and a traditionally challenging area of mathematical analysis.

In high-dimensional spaces, PINNs minimize the integrated residual of a PDE via SGD to find a continuously differentiable function that approximately satisfies the PDE. When applying this idea to solve a CDE, an integral equation, the solving procedure becomes more *natural* in the following two ways. First, we only assume that the CDE solution is Lipschitz continuous, and neural networks are inherently Lipschitz continuous. Second, as SGD is already used to handle the integrated residual, we can simultaneously use it to handle the integral in the CDE. After approximately solving the CDE, DCDC needs to convert the solution into a convergence bound, which requires that the solution is uniformly accurate with high probability. This is different from PINNs in the PDE literature since the accuracy is mainly measured in the $L_2$ sense.

The CDE solution is a new type of Lyapunov function that provides explicit convergence rates for random dynamical systems. For deterministic dynamical systems, traditional Lyapunov functions play central roles in establishing stability; see (Pukdeboon, 2011) for a review. There is a substantial literature on computing traditional Lyapunov functions via neural networks; see (Liu et al., 2023) and references therein. As pointed out in (Dawson et al., 2023), a survey on certificate learning, learned (traditional) Lyapunov functions provide safety certificates for learned control policies (on deterministic dynamical systems). For the control of random dynamical systems, DCDC not only generates safety certificates (CDE solutions) but also quantifies safety levels (convergence rates). Control and performance evaluation of random dynamical systems have become a staple in contemporary data-driven decision making systems, thus underscoring the importance of DCDC.

In short, we summarize our contributions as follows:

- We introduce the Deep Contractive Drift Calculator (DCDC), the first general-purpose end-to-end approach that enables the use of deep learning to bound the convergence rate of general state-space Markov chains.

- We perform sample complexity analysis and use DCDC to generate convergence bounds for realistic Markov chains arising in operations research as well as machine learning.
- Our DCDC approach discovers features that are exploited by techniques developed to study CDs by closed-form methods, such as the wedge shape and the boundary removal technique discussed in (Qu et al., 2023).

## 2   Contractive Drift Equation

Let $X$ be a Markov chain on $\mathcal{X} \subset \mathbb{R}^d$, with random mapping representation

$$X_{n+1} = f_{n+1}(X_n), \ \ n = 0, 1, 2, \ldots$$

where $f_n$'s are iid copies of $f$, a locally Lipschitz random mapping from $\mathcal{X}$ to itself (with probability one, $f : \mathcal{X} \to \mathcal{X}$ is locally Lipschitz).

**Example.** Let $\alpha$ be a positive constant and $Z$ be a square integrable random variable. The SGD with step-size $\alpha$ to solve $\min_x \mathbb{E}(x - Z)^2/2$ is $X_{n+1} = X_n - \alpha(X_n - Z_{n+1})$ where $Z_{n+1}$'s are iid copies of $Z$, so the corresponding random mapping is $f(x) = x - \alpha(x - Z)$.

Understanding the long-term behavior of $X$ requires estimating how fast $X_n$ converges to $X_\infty$ (equilibrium) as $n \to \infty$. The difference between the two distributions is quantified by either total variation (TV) distance or Wasserstein distance. Representative TV convergence bounds (e.g., $TV(X_n, X_\infty) \leq Cr^n$) can be found in (Meyn et al., 1994; Rosenthal, 1995; Baxendale, 2005). Representative Wasserstein convergence bounds (e.g., $W(X_n, X_\infty) \leq Cr^n$) can be found in (Hairer et al., 2011; Durmus and Moulines, 2015; Qin and Hobert, 2022a). These analytical methods can only handle stylized (structured) Markov chains. The goal of this paper is to introduce the first computational method that can handle realistic (less structured) Markov chains.

The first step to achieve the goal is introducing the contractive drift equation (CDE). The local Lipschitz constant of $f$ at $x \in \mathcal{X}$ is defined as

$$Df(x) \triangleq \lim_{\delta \to 0} \sup_{x', x'' \in B_\delta(x)} \frac{\|f(x') - f(x'')\|}{\|x' - x''\|}$$

where $\|\cdot\|$ is the Euclidean norm and $B_\delta(x) = \{x' : \|x' - x\| < \delta\}$. If $f$ is differentiable, then

$$Df(x) = \lim_{h \to 0} \sup_{v: \|v\|=1} \frac{\|f(x + hv) - f(x)\|}{h} = \sup_{v: \|v\|=1} \|\nabla f(x)v\| = \|\nabla f(x)\|$$

where $\nabla f$ is the Jacobian matrix of $f$ and $\|\cdot\|$ becomes the spectral norm when applying to matrices. Basically, $Df(x)$ describes how expansive or contractive $f$ is around $x$. With these notations, the contractive drift condition (CD) in (Qu et al., 2023) that leads to computable convergence bounds is

$$KV(x) \triangleq \mathbb{E}Df(x)V(f(x)) \leq V(x) - U(x), \ \ x \in \mathcal{X} \tag{1}$$

where $V, U : \mathcal{X} \to \mathbb{R}_+$ are bounded away from zero. In the rest of this paper, we adopt the convention that all functions denoted by $U$ are positive and bounded away from zero, i.e. $\inf U > 0$. We use $\mathbb{E}_x$ to denote the expectation operator conditional on $X_0 = x$. In (1), the subscript is omitted as the initial location is clear. By replacing "$\leq$" with "$=$" in (1), the contractive drift equation (CDE) is $KV = V - U$, for which we establish the following existence and uniqueness results. All proofs are in the appendix.

**Theorem 1.** *Fix $U$ and suppose that $KW \leq W - U$ has a non-negative finite solution $W_*$. Then*

$$V_*(x) \triangleq \mathbb{E}_x \left[ \sum_{k=0}^{\infty} U(X_k) \prod_{l=1}^{k} Df_l(X_{l-1}) \right], \ \ x \in \mathcal{X} \tag{2}$$

*is finite and satisfies $KV_* = V_* - U$. Furthermore, $KV = V - U$ has at most one bounded solution.*

*Remark.* This $V_*$ can be interpreted as an average space-discounted cumulative reward. Imagine a swarm of agents moving according to $f$. For an agent at $x$, if $Df(x) < 1$ (contraction), then after $f$ is applied, there will be more agents around this agent. If all agents around $f(x)$ share a total reward $U(f(x))$, then the reward for each of them is discounted. From the perspective of a particular agent, the procedure is like collecting reward within a shrinking ball.

# 3 Deep Contractive Drift Calculator

## 3.1 Why do we introduce CDE?

Physics-informed neural networks (PINNs) solve a PDE by minimizing its integrated residual (Sirignano and Spiliopoulos, 2018; Raissi et al., 2019). If we want to use this idea to solve $KV \leq V - U$, then the integrated residual is

$$\bar{l}(\theta) \triangleq \int_{\mathcal{X}} (KV_\theta(x) - V_\theta(x) + U(x))_+ \, h(x)dx$$

where $h$ is a positive density and $\{V_\theta : \theta \in \Theta\}$ is a neural network. Note that the residual at $x$ is positive if and only if $KV_\theta(x) > V_\theta(x) - U(x)$. By letting $X_0$ have distribution $h$,

$$\bar{l}(\theta) = \mathbb{E}\left[\mathbb{E}\left[Df_1(X_0)V_\theta(f_1(X_0)) - V_\theta(X_0) + U(X_0)|X_0\right]\right]_+,$$

which is an expectation of a non-linear function of a conditional expectation. Minimizing $\bar{l}(\theta)$ is a conditional stochastic optimization problem (CSO). In CSO, the sample-average gradient is biased (Hu et al., 2020b), which leads to a high sample complexity for convergence (Hu et al., 2020a). Fortunately, if we aim at solving $KV = V - U$ (CDE) instead of $KV \leq V - U$ (CD), then there exists a simple unbiased gradient estimator. Now we briefly derive this estimator. For a CDE, the integrated residual becomes

$$l(\theta) \triangleq \int_{\mathcal{X}} (KV_\theta(x) - V_\theta(x) + U(x))^2 \, h(x)dx$$

$$= \mathbb{E}\left[\mathbb{E}\left[Df_1(X_0)V_\theta(f_1(X_0)) - V_\theta(X_0) + U(X_0)|X_0\right]\right]^2$$

with its gradient

$$l'(\theta)$$
$$= 2\mathbb{E}\left[\mathbb{E}\left[Df_1(X_0)V_\theta(f_1(X_0)) - V_\theta(X_0) + U(X_0)|X_0\right]\mathbb{E}\left[Df_1(X_0)V_\theta'(f_1(X_0)) - V_\theta'(X_0)|X_0\right]\right]$$
$$= 2\mathbb{E}\mathbb{E}\left[\left[Df_1(X_0)V_\theta(f_1(X_0)) - V_\theta(X_0) + U(X_0)\right]\left[Df_{-1}(X_0)V_\theta'(f_{-1}(X_0)) - V_\theta'(X_0)\right]|X_0\right]$$
$$= 2\mathbb{E}\left[\left[Df_1(X_0)V_\theta(f_1(X_0)) - V_\theta(X_0) + U(X_0)\right]\left[Df_{-1}(X_0)V_\theta'(f_{-1}(X_0)) - V_\theta'(X_0)\right]\right]$$

where $f_1$ and $f_{-1}$ are iid copies of $f$ while $V_\theta' = dV_\theta/d\theta$ is computed via backpropagation. This expression allows us to estimate $l'(\theta)$ without any bias. In summary, the inequality (CD) is enough to bound the convergence, but the equality (CDE) turns out to be easier to establish (via deep learning).

## 3.2 DCDC

Given the above discussion, a standard application of SGD is enough to simultaneously handle the integrated residual as well as the integral in the CDE, resulting in the following simple algorithm, Deep Contractive Drift Calculator, the first general-purpose sample-based algorithm to bound the convergence of general state-space Markov chains.

---
**Algorithm 1** Deep Contractive Drift Calculator (DCDC)

---
**Require:** Step-size $\alpha$, number of iterations $T$, neural network $\{V_\theta : \theta \in \Theta\}$, initialization $\theta_0$
    **for** $t \in \{0, ..., T-1\}$ **do**
        sample $(X_0, f_1, f_{-1})$
        compute $\hat{l}'(\theta_t)$ as

$$2\left[Df_1(X_0)V_{\theta_t}(f_1(X_0)) - V_{\theta_t}(X_0) + U(X_0)\right]\left[Df_{-1}(X_0)V_{\theta_t}'(f_{-1}(X_0)) - V_{\theta_t}'(X_0)\right]$$

        update $\theta_{t+1} = \theta_t - \alpha\hat{l}'(\theta_t)$ (SGD or its variants)
    **end for**
    convert $V_{\theta_T}$ into a convergence bound (Theorem 3 and Theorem 4)

---

The conversion will be discussed in the next two subsections. In the current subsection, we show the validity of approximating CDE solutions via neural networks. In the following, we use $\|\cdot\|_\infty$ to denote the sup norm of functions on $\mathcal{X}$.

**Theorem 2.** *If $\mathcal{X}$ is compact, $\|\mathbb{E}Df\|_\infty$ is finite, and $V_*$ in (2) is finite and continuous, then for any $\epsilon > 0$, there exists a neural network $\{V_\theta : \theta \in \Theta\}$ and its realization $V_{\theta_*}$ such that*

$$\|KV_{\theta_*} - V_{\theta_*} + U\|_\infty < \epsilon.$$

Although DCDC solves CDEs on compact sets, it can be applied to Markov chains on non-compact sets that have compact absorbing sets (e.g. SGD for regularized problems). For chains without a compact absorbing set, extending DCDC to bound their convergence is left for future research, but here we describe a natural strategy to do so. In general, a Markov chain spends most of its time on some large compact set $C$ where the chain may have complex dynamics. When the chain is outside $C$, it typically has a strong tendency to return. Therefore, to extend DCDC, we can (i) search some parametric family (e.g. $V_A(x) = x^\top A x$) to establish a CD outside $C$ (capturing the return tendency); (ii) apply DCDC to obtain a CDE solution on $C$ (capturing the complex dynamics); (iii) stitch them together to obtain a global CD. Comparing the large set here with the *small set* (Meyn and Tweedie, 2009) in D&M or D&C illustrates the advantage of computational methods over analytical ones. The size of the large set is determined by the approximation capability of neural networks, but the size of the small set is determined by the minorization or contraction condition (the two conditions often require the small set to be very small).

## 3.3 Practical convergence bounds with exponential rates

Now we discuss how to convert $KV \leq V - U$ into convergence bounds with exponential rates in Wasserstein distance. To begin, we recall the definition of the Wasserstein distance. Let $\mathcal{P}(\mathcal{X})$ be the set of probability measures on $\mathcal{X}$ equipped with its Borel sigma-algebra. The Wasserstein distance between $\mu, \nu \in \mathcal{P}(\mathcal{X})$ is

$$W(\mu, \nu) \triangleq \inf_{\pi \in \mathcal{C}(\mu,\nu)} \int_{\mathcal{X} \times \mathcal{X}} \|x - y\| \, \pi(dx, dy)$$

where

$$\mathcal{C}(\mu, \nu) \triangleq \{\pi \in \mathcal{P}(\mathcal{X} \times \mathcal{X}) : \pi(\cdot, \mathcal{X}) = \mu(\cdot), \ \pi(\mathcal{X}, \cdot) = \nu(\cdot)\}$$

is the set of all couplings of $\mu$ and $\nu$. Given two random variables $Z_1$ and $Z_2$, we use $W(Z_1, Z_2)$ to denote the Wasserstein distance between their marginal distributions.

**Theorem 3.** *Suppose that $\mathcal{X}$ is convex and that $KV \leq V - U$ holds with $\sup V < \infty$. If $\mathbb{E}\|X_0 - X_1\| < \infty$, then $X$ has a unique stationary distribution $X_\infty$ with*

$$W(X_n, X_\infty) \leq Cr^n, \quad r \triangleq 1 - \inf U / \sup V, \quad C \triangleq \frac{\mathbb{E}\|X_0 - X_1\| \, V(X_0 + \tilde{U}(X_1 - X_0))}{\inf U \cdot (\inf V / \sup V)}$$

*where $\tilde{U}$ is a $U[0, 1]$ random variable independent of $X_0$ and $X_1$.*

Given $U$, the exponential rate $r$ is determined by the magnitude of $V$. The smaller the $V$, the faster the convergence. Given $X_0$, the pre-multiplier $C$ can be easily computed by simulating the first transition (from $X_0$ to $X_1$).

In Theorem 3 of (Qu et al., 2023), convergence bounds with exponential rates are straightforwardly derived from $KV \leq rV$ where $r < 1$, so one might wonder why we need the less straightforward Theorem 3 here. This is because $KV \leq rV$ is not suitable for PINN-like solvers. In Theorem 3, we solve $KV = V - U$ and compute the exponential rate $r$ from the solution $V$. However, for $KV = rV$, we need the answer (the exponential rate $r$) to write down the question (the equation to solve and the corresponding loss to minimize), which is circular. Of course, we may try solving $KV = rV$ for different values of $r$, but it turns out that it is very hard for DCDC to converge even for very conservative (close to 1) $r$'s. Here is an explanation. Unlike $KV = V - U$, which has a solution as long as $KV \leq V - U$ has one (Theorem 1), $KV = rV$ may not have a solution even when $KV \leq rV$ has one. However, it is not hard to show that $KV = rV - r$ has a (formal) solution

$$V_r(x) \triangleq \mathbb{E}_x\left[\sum_{k=0}^\infty (1/r)^k \prod_{l=1}^k Df_l(X_{l-1})\right], \quad x \in \mathcal{X}.$$

Comparing with $V_*$ in (2), $U(X_k)$ is replaced by exponentially exploding $(1/r)^k$. Back to $KV = rV$, its solution (if there is any) should be the above expression without the summation but with $k \to \infty$ (as a limit), which suggests that the solution may have a large magnitude, making it difficult to approximate.

### 3.4 Practical convergence bounds with polynomial rates

Now we discuss how to generate convergence bounds with polynomial rates using DCDC. The key is to iteratively solve a sequence of CDEs. For example, given $V_0$, we first solve $KV_1 = V_1 - V_0$ to obtain $V_1$. Then we solve $KV_2 = V_2 - V_1$ to obtain $V_2$. These two CDEs together lead to an $O(1/n)$ convergence bound.

**Theorem 4.** *Suppose that $\mathcal{X}$ is convex and that there exist positive functions $V_0, V_1, \ldots, V_m$ such that $0 < \inf V_0 < \sup V_m < \infty$ and $KV_{k+1} \leq V_{k+1} - V_k$ for $k = 0, \ldots, m-1$. If $\mathbb{E}\,\|X_0 - X_1\| < \infty$, then $X$ has a unique stationary distribution $X_\infty$ with*

$$W(X_n, X_\infty) \leq \frac{\mathbb{E}\,\|X_0 - X_1\|\, V_m(X_0 + \tilde{U}(X_1 - X_0))}{\inf V_0 \cdot \prod_{k=1}^{m-1}\left(1 + n/k\right)}$$

*where $\tilde{U}$ is a $U[0,1]$ random variable independent of $X_0$ and $X_1$.*

The expectation in the numerator can be easily computed by simulating the first transition, while the product in the denominator is basically $n^{m-1}$ as $n \to \infty$.

In Theorem 1 of (Qu et al., 2023), convergence bounds with polynomial rates ($O(1/n^{m-1})$) are derived from $KV \leq V - U^{1/m}V^{1-1/m}$ paired with $KU \leq U$, so one might wonder why we need so many CDs in Theorem 4 here. This is because $KV \leq V - U^{1/m}V^{1-1/m}$ is designed for the pen-and-paper setting where directly establishing a sequence of CDs is difficult. Given $KV \leq V - U^{1/m}V^{1-1/m}$, many inequalities are applied to extract a CD sequence from this single special CD, resulting in large constants in convergence bounds. DCDC makes it possible to directly establish a sequence of CDs (by consecutively solving CDEs). In this setting, we can use Theorem 4 to obtain better convergence bounds. To be specific, compared with our Theorem 4, the result in (Qu et al., 2023) has an extra factor $m^m/m!$.

## 4 Sample Complexity

As a numerical solver, DCDC solves CDEs approximately. Let $\tilde{V} = V_{\theta_T}$ be the output of DCDC. We should not expect $K\tilde{V} = \tilde{V} - U$ to hold exactly. Even if $\tilde{V}$ is an exact solution, the exactness is hard to verify as $K\tilde{V}$ is an expectation and the domain $\mathcal{X} \subset \mathbb{R}^d$ is not a finite set. As establishing convergence bounds requires CDs to exactly hold everywhere, given $N$ iid copies of $f$ to estimate $K$ and $\mathcal{M} = \{x_1, \ldots, x_M\}$ uniformly sampled from $\mathcal{X}$, we can (i) establish

$$\hat{K}_N\tilde{V}(x) \triangleq \frac{1}{N}\sum_{k=1}^{N} Df_k(x)\tilde{V}(f_k(x)) \leq \tilde{V}(x) - \tilde{U}(x), \quad x \in \mathcal{M}$$

where $\tilde{U}$ may be smaller than $U$ (e.g. if $\tilde{V}$ is supposed to solve $\tilde{V} - K\tilde{V} = U \equiv 1$, then $\tilde{U} \equiv \inf_{\mathcal{M}}[\tilde{V} - \hat{K}_N\tilde{V}]$); (ii) claim that $K\tilde{V} \leq \tilde{V} - \tilde{U} + \epsilon$ holds everywhere with probability at least $1 - \delta$ where $\epsilon, \delta > 0$; (iii) convert $K\tilde{V} \leq \tilde{V} - \tilde{U} + \epsilon$ into a convergence bound. To have $M, N$ large enough to make the claim in (ii), we need the following sample complexity result.

**Theorem 5.** *Suppose that (i) $\mathcal{X}$ is compact; (ii) $V, U$ are bounded and Lipschitz; (iii) $\mathbb{E}Df^2 + \mathbb{E}D^2f < \infty$ where $Df$ is the Lipschitz constant of $f$ and $D^2f$ is the Lipschitz constant of $Df$. Given $\epsilon, \delta > 0$, we can choose $M = O(\log(1/\epsilon)/(\delta\epsilon^d))$ and $N = O(1/(\delta\epsilon^2))$ to have*

$$P\left(\sup_{x \in \mathcal{X}}\left[KV(x) - V(x) + U(x)\right] \leq \sup_{x \in \mathcal{M}}\left[\hat{K}_N V(x) - V(x) + U(x)\right] + \epsilon\right) > 1 - \delta.$$

Since the exponential rate of convergence in Theorem 3 is $r = 1 - \inf U/\inf V$, Theorem 5 also provides the sample complexity for estimating the exponential rate. Specifically, with probability at least $1 - \delta$, the exponential rate $\hat{r}_{M,N}$ computed from $\hat{K}_N V \leq V - U$ on $\mathcal{M}$, which may not be a valid exponential rate, is $\epsilon$-close to a valid exponential rate $r_*$ (given by $KV \leq V - U + \epsilon$ on $\mathcal{X}$).

It is worth noting that in terms of sample complexity, Theorem 5 guarantees a DCDC certificate (and thus a convergence bound to stationarity) with high probability with an efficient parametric $O(1/N^{1/2})$ rate in terms of the number of samples (namely, the bound holds with high probability

up to an error of order $O(1/N^{1/2})$). Once samples are generated, $M = \tilde{O}(1/\epsilon^d)$ points are chosen for the empirical evaluation. Thus, the total complexity (both in terms of number of evaluations and number of samples is $O(1/\epsilon^2) + \tilde{O}(1/\epsilon^d)$. A related literature on parametric integration (i.e. learning a Markov transition kernel that maps Lipschitz functions to continuous functions on the $d$-dimensional cube) provides a lower bound of order $\tilde{O}(1/\epsilon^d)$, (Heinrich and Sindambiwe, 1999). Although these results are suggestive, they cannot be applied directly because we assume a random mapping representation, which provides additional structure. We plan to study the lower bounds in future work.

## 5 Numerical Examples

### 5.1 Mini-batch SGD for logistic regression with regularization

Having established the theoretical foundation of DCDC, we now utilize it to generate convergence bounds for Markov chains of interest that are too hard for pen-and-paper analysis. To begin, we bound the convergence of a constant step-size mini-batch SGD that minimizes the cross-entropy loss over a finite dataset with $L_2$ regularization.

Let $(x_1, y_1), \ldots, (x_m, y_m)$ be $m$ data points where $x_i \in [-1/2, 1/2]^2$ and $y_i \in \{0, 1\}$. To perform regularized logistic regression, we want to choose $b \in \mathbb{R}^2$ to minimize

$$-\frac{1}{m} \sum_{i=1}^{m} (y_i \log p_i + (1 - y_i) \log(1 - p_i)) + \frac{\lambda}{2m} \|b\|^2, \quad p_i = \sigma(b^\top x_i) = \frac{1}{1 + \exp(-b^\top x_i)}$$

where $\lambda > 0$ is the regularization parameter. The random mapping representation of the corresponding SGD with step-size $\alpha$ and batch-size $\beta$ is

$$f(b) = b(1 - \lambda\alpha/m) + (\alpha/\beta) \sum_{i \in B} \left[y_i - \sigma(b^\top x_i)\right] x_i$$

where $B$ with $|B| = \beta$ is uniformly sampled from $\{1, \ldots, m\}$. Thanks to the regularization, the chain has a compact absorbing set. In fact, the chain cannot escape from $B_{m/(\lambda\sqrt{2})}(0)$. The local Lipschitz constant of $f(b)$ is

$$Df(b) = \left\| (1 - \lambda\alpha/m)I - (\alpha/\beta) \sum_{i \in B} \sigma'(b^\top x_i) x_i x_i^\top \right\| \leq 1 - \lambda\alpha/m$$

where $\|A\| = \sup_{v: \|v\|=1} \|Av\|$ is the spectral norm. This demonstrates that the $L_2$ regularization makes the chain contractive $\|f(b_1) - f(b_2)\| \leq (1 - \lambda\alpha/m) \|b_1 - b_2\|$. However, since the regularization parameter is chosen via cross-validation in a separate validation process, it is useful to obtain a contraction rate that is uniform in the regularization parameter. This rate is brought by the second term in $Df(b)$ - we refer to this contribution as the *intrinsic* convergence rate. However, it is challenging to analyze the spectrum of this state-dependent data-based random matrix, so we need DCDC. The code is available in the supplementary material. Each training procedure in this paper was completed within ten minutes on an M2 MacBook Air with 8GB RAM.

For the dataset, we set $m = 100$ and uniformly generate 100 $x_i$'s. For each $x_i$, its label $y_i$ follows $\text{Ber}(0.9)$ or $\text{Ber}(0.1)$, depending upon which coordinate of $x_i$ is larger. For the SGD, we set the regularization parameter $\lambda = 1$, step-size $\alpha = 0.1$, and batch-size $\beta = 10$. For DCDC, we run 1M Adam steps to train a single-layer network with width 1000 and sigmoid activation. We also experiment with deeper networks with the same amount of neurons, and the results are similar.

As demonstrated in Figure 5.1, the single-layer network can already accurately solve the CDE $KV = V - 0.1$. The learned solution $\tilde{V}$ is on the left while the estimated difference $\hat{K}\tilde{V} - \tilde{V}$ is on the right. Aiming at $KV \leq V - 0.1$, we get $\hat{K}\tilde{V} \leq \tilde{V} - 0.0986$. This leads to exponential rate $1 - 1.07 \times 10^{-3}$ (Theorem 3) where $1 \times 10^{-3}$ corresponds to the regularization contribution, while $7 \times 10^{-5}$ corresponds to the intrinsic rate. Now we briefly discuss how the surface in Figure 5.1 (left) leads to the intrinsic rate. From the expression of $Df(b)$, we know that the intrinsic contraction concentrates around the center. To make it contribute to the overall convergence, it needs to be *spread*. The surface in Figure 5.1 (left) provides the media to spread: (i) for points not

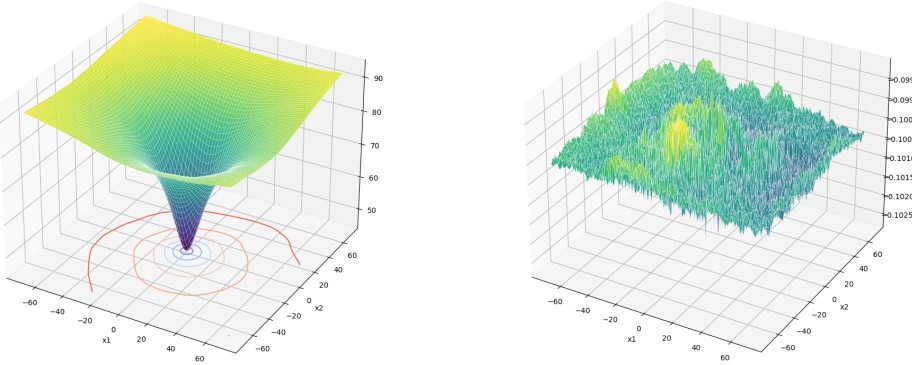

Figure 1: Left: The learned solution $\tilde{V}$ of $KV - V = -0.1$ for the mini-batch SGD, with maximum 91.87. Right: The estimated difference $\hat{K}\tilde{V} - \tilde{V}$, with maximum -0.0986, mean -0.9999, standard deviation 0.0003.

at the center, the sunken surface creates a drift $\mathbb{E}_x V(X_1) < V(x)$; (ii) however, for points at the center, the sunken surface creates an anti-drift $\mathbb{E}_x V(X_1) > V(x)$, but it is overcome by the strong contraction $\mathbb{E}_x Df_1(x)V(X_1) < V(x)$. In this way, the strong contraction is spread (in the form of drift) to overall improve the contractive drift, which leads to the intrinsic rate. To conclude this example, when $X_0 = 0$, we compute the pre-multiplier $C = 8.1$, which leads to convergence bound $W(X_n, X_\infty) \leq 8.1(1 - 1.07 \times 10^{-3})^n$.

## 5.2 Tandem fluid networks

In the above SGD example, contraction plays the leading role. Now we consider a tandem fluid network (Kella and Whitt, 1992) where drift plays the leading role. Let $s_1$ and $s_2$ be two stations with buffer capacity $c$ that can process fluid workload at rates $r_1$ and $r_2$, respectively. External fluid only arrives at $s_1$ and is processed by $s_1$ then $s_2$. Assume that the external input follows a compound renewal process where a random amount of fluid $Z$ arrives after a random length of time $T$ has passed since the last arrival. If $r_1 < r_2$, then $s_2$ is always empty, so we let $r_1 > r_2$. Let $X$ be the remaining workload vector after each arrival. Its random mapping representation is

$$f(x_1, x_2) = (\min((x_1 - r_1 T)_+ + Z, c), (\min(x_2 + (r_1 - r_2)\min(T, x_1/r_1), c) - r_2(T - x_1/r_1)_+)_+)$$

where $x_1$ decreases at rate $r_1$ until it is empty while $x_2$ increases at rate $(r_1 - r_2)$ until $x_1$ is empty. Basically, within $[0, c]^2$, the chain follows a northwest-then-south path for time $T$ and then jumps east by amount $Z$. This chain has simple local Lipschitz constant $Df(x_1, x_2) = I(T \leq (x_1 + x_2)/r_2)$, obtained as an infinitesimal ball around $(x_1, x_2)$ collapses to a single point when the system is depleted before the next arrival. As a result, drift plays the leading role as contraction only happens around the origin.

For the tandem network, we set the buffer capacity $c = 1$, processing rates $(r_1, r_2) = (1.1, 1.0)$, interarrival time $T \sim U[0, 0.2]$ and arriving amount $Z \sim U[0, 0.1]$ (the stability condition is $\mathbb{E}Z < r_2\mathbb{E}T$). For DCDC, we run 1M Adam steps to train a double-layer network with width 40 and sigmoid activation.

Although a slightly deeper network is trained, the result in Figure 5.2 (left) is almost a plane. Now we briefly explain why this is the correct solution. First, note that as long as the stability condition $\mathbb{E}Z < r_2\mathbb{E}T$ holds, the total workload $\bar{V}(x_1, x_2) = x_1 + x_2$ is the most natural Lyapunov function such that $\mathbb{E}_x\bar{V}(X_1) - \bar{V}(x) = \mathbb{E}(-r_2 T + Z) < 0$ holds when $x$ is far away from the boundary. Second, the "boundary removal technique" introduced in (Qu et al., 2023) shows that the above drift can be extended to the boundary as a contractive drift $\mathbb{E}_x Df(x)\bar{V}(X_1) - \bar{V}(x) = \mathbb{E}(-r_2 T + Z) < 0$ as if the boundary (that causes anti-drift) never exists. The plane in Figure 5.2 (left) demonstrates that DCDC has already mastered the above two steps! Again, we conclude this example with convergence bound $W(X_n, X_\infty) \leq 5.67(1 - 0.017)^n$ when $X_0 = 0$.

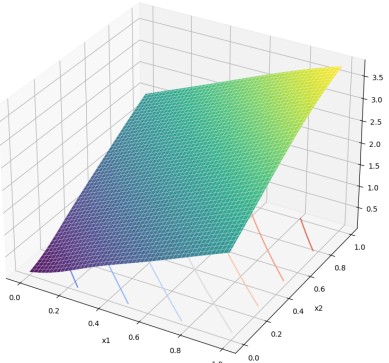 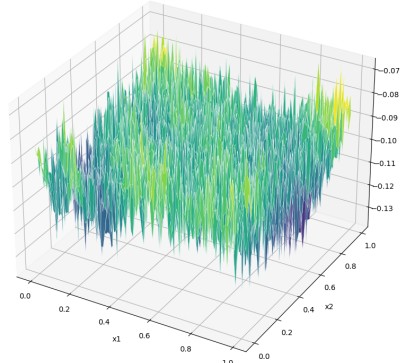

Figure 2: Left: The learned solution $\tilde{V}$ of $KV - V = -0.1$ for the tandem network, with maximum 3.78. Right: The estimated difference $\hat{K}\tilde{V} - \tilde{V}$, with maximum -0.0668, mean -0.0989, standard deviation 0.0097.

### 5.3 Discovery of meaningful wedge-like Lyapunov functions

Lyapunov functions are usually denoted by $V$ in the literature, and $V$ typically represents the shape of Lyapunov functions. As mentioned in the introduction, the CDE solution is a new type of Lyapunov function. In most cases, it is also $V$-shaped, representing the drift towards some contractive region. However, Markov chains sometimes exhibit neither drift nor contraction, such as when SGD is stuck in a non-strongly-convex basin or when the water level of the Moran dam (Stadje, 1993) is neither too low nor too high. Here, we use the simplest example, a two-sided regulated random walk $f(x) = \max(\min(x + Z, 1/2), -1/2)$ with $Z \sim U[-1/3, 1/3]$, to illustrate that DCDC discovers upside-down $\Lambda$-shaped Lyapunov functions to address the above issue.

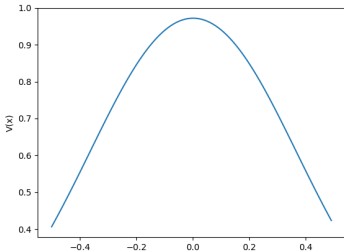 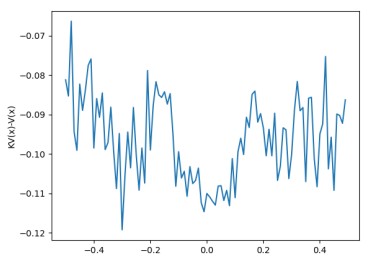

Figure 3: Left: The learned solution $\tilde{V}$ of $KV - V = -0.1$ for the regulated random walk, with maximum 0.972. Right: The estimated difference $\hat{K}\tilde{V} - \tilde{V}$, with maximum -0.0662, mean -0.0964, standard deviation 0.0106.

In $[-1/6, 1/6]$, the chain exhibits neither drift ($Z$ is symmetric) nor contraction ($\pm 1/2$ boundaries are not reachable in one step). The wedge in Figure 5.3 (left) creates an artificial drift to maintain the CD. In (Qu et al., 2023), a similar function is introduced as a tool to study stylized non-strongly-convex SGD. DCDC not only discovers this tool but also makes the wedge meaningful. As mentioned in the remark below Theorem 1, the CDE solution generated by DCDC represents an average space-discounted cumulative reward, where an agent collects reward within a shrinking ball. Why does starting from the middle lead to the highest reward? Because the ball starting there has the longest lifespan before hitting the boundary and collapsing into a single point.

## 5.4 Recovery of exact convergence rates

Finally, to demonstrate the potential of DCDC to accurately recover exact convergence rates, we examine a class of $d$-dimensional autoregressive processes

$$f(x) = Hx + Z, \quad x, H, Z \geq 0 \qquad (3)$$

where random vector $Z$ is integrable and constant matrix $H$ is symmetric with all its eigenvalues in $(0, 1)$. The exact convergence rate of this Markov chain in Wasserstein distance can be explicitly computed by pen and paper.

**Proposition 1.** *Let $X$ be the Markov chain defined by* (3). *If $X_0 = 0$, then*

$$\mathbb{E} \|H^n Y\| / \sqrt{d} \leq W(X_n, X_\infty) \leq \mathbb{E} \|H^n Y\|, \quad Y = \sum_{k=1}^{\infty} H^{k-1} Z_k$$

*where $Z_k$'s are iid copies of $Z$. Let $\lambda$ be the largest eigenvalue of $H$. Then $\mathbb{E} \|H^n Y\| = \Theta(\lambda^n)$ as long as $Y$ does not concentrate on the orthogonal complement of the eigenspace associated with $\lambda$.*

For the autoregressive process, let $d = 3$ and

$$H = \begin{pmatrix} 0.4 & 0.2 & 0.1 \\ 0.2 & 0.5 & 0.2 \\ 0.1 & 0.2 & 0.6 \end{pmatrix}$$

with $\lambda = 0.850$. Let $Z$ be uniformly sampled from $B_1(0) \cap \mathbb{R}^3_+$. Note that the resulting Markov chain cannot escape from $B_{10}(0) \cap \mathbb{R}^3_+$. After plugging into the simulator of (3), DCDC generates $V \equiv 0.668$, which recovers the exact convergence rate

$$KV = V - U = (1 - U/V)V = (1 - 0.1/0.668)V = 0.850V = \lambda V.$$

## 6 Conclusions

We introduce DCDC, a potent framework that enables the use of deep learning techniques to tackle the problem of estimating convergence to stationarity of complex, general state-space Markov chains. Our approach unlocks the key to using scalable data-driven tools to tackle this important problem. In future work, we plan to use these results in the context of general state-space reinforcement learning, control of ergodic systems, and related applications by employing the CD condition as a policy regularizer.

## 7 Limitations

DCDC solves CDEs on compact spaces. A potential strategy to handle non-compact spaces is discussed in Section 3.2. Sample complexity lower/upper bounds are not studied in this paper and they are left for future research.

## Acknowledgments and Disclosure of Funding

The material in this paper is based upon work supported by the Air Force Office of Scientific Research under award number FA9550-20-1-0397. Additional support is gratefully acknowledged from NSF 2118199, 2229012, 2312204, and ONR 13983111.

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

## A  Appendix

*Proof of Theorem 1.* Note that

$$
\begin{aligned}
W_*(x) \geq & KW_*(x) + U(x) \\
= & \mathbb{E}Df_1(x)W_*(f_1(x)) + U(x) \\
\geq & \mathbb{E}Df_1(x)(KW_*(f_1(x)) + U(f_1(x))) + U(x) \\
= & \mathbb{E}Df_1(x)\mathbb{E}\left[Df_2(f_1(x))W_*(f_2(f_1(x)))|f_1\right] + \mathbb{E}Df_1(x)U(f_1(x)) + U(x) \\
= & \mathbb{E}Df_1(x)Df_2(f_1(x))W_*(f_2(f_1(x))) + \mathbb{E}Df_1(x)U(f_1(x)) + U(x) \\
& \cdots \\
\geq & \mathbb{E}_x\left[W_*(X_n)\prod_{k=1}^{n}Df_k(X_{k-1})\right] + \sum_{k=0}^{n-1}\mathbb{E}_x\left[U(X_k)\prod_{l=1}^{k}Df_l(X_{l-1})\right] \\
\geq & \sum_{k=0}^{n-1}\mathbb{E}_x\left[U(X_k)\prod_{l=1}^{k}Df_l(X_{l-1})\right].
\end{aligned}
$$

As $n \to \infty$, we have $V_* \leq W_* < \infty$ and

$$
\begin{aligned}
KV_*(x) &= \mathbb{E}Df_1(x)\mathbb{E}_{X_1}\left[\sum_{k=0}^{\infty}U(X_{k+1})\prod_{l=1}^{k}Df_{l+1}(X_l)\right] \\
&= \mathbb{E}\left[\sum_{k=0}^{\infty}U(X_{k+1})\prod_{l=0}^{k}Df_{l+1}(X_l)\right] \\
&= \mathbb{E}\left[\sum_{k=1}^{\infty}U(X_k)\prod_{l=1}^{k}Df_l(X_{l-1})\right] \\
&= V_*(x) - U(x).
\end{aligned}
$$

Let $V^*$ be another solution of $KV = V - U$. Similar to $W_*$,

$$
V^*(x) = \mathbb{E}_x\left[V^*(X_n)\prod_{k=1}^{n}Df_k(X_{k-1})\right] + \sum_{k=0}^{n-1}\mathbb{E}_x\left[U(X_k)\prod_{l=1}^{k}Df_l(X_{l-1})\right]. \tag{4}
$$

As $n \to \infty$, we have $V^* \geq V_*$. If $V_*$, and hence $V^*$, is unbounded, then $KV = V - U$ doesn't have bounded solution. If $V^*$, and hence $V_*$, is bounded, we claim that they are the same solution. It suffices to show that the first term on the RHS of (4) vanishes as $n \to \infty$. This is true because

$$
V_*(x) < \infty \;\Rightarrow\; \mathbb{E}_x\left[U(X_n)\prod_{k=1}^{n}Df_k(X_{k-1})\right] \to 0 \;\Rightarrow\; \mathbb{E}_x\left[V^*(X_n)\prod_{k=1}^{n}Df_k(X_{k-1})\right] \to 0
$$

where the last step follows from $\sup V^* < \infty$ and $\inf U > 0$. $\qquad\square$

*Proof of Theorem 2.* Since continuous functions on compacts sets are bounded, by Theorem 1, $V_*$ is the unique continuous solution of $KV = V - U$. By the universal approximation theorem (Cybenko, 1989), there exists a single-layer neural network with sigmoid activation $\{V_\theta : \theta \in \Theta\}$ ($\Theta$ is some Euclidean space) and its realization $V_{\theta_*}$ such that $\|V_{\theta_*} - V_*\|_\infty < \epsilon/(\bar{L}+1)$ where $\bar{L} = \|\mathbb{E}Df\|_\infty$. Then

$$
\begin{aligned}
\|KV_{\theta_*} - V_{\theta_*} + U\|_\infty &\leq \|KV_* - V_* + U\|_\infty + \|V_{\theta_*} - V_*\|_\infty + \|KV_{\theta_*} - KV_*\|_\infty \\
&< 0 + \epsilon/(\bar{L}+1) + \bar{L}\epsilon/(\bar{L}+1) \\
&= \epsilon.
\end{aligned}
$$

$\qquad\square$

*Proof of Theorem 3.* The setting introduced in Section 2 is a special case of the setting in (Qu et al., 2023), allowing us to directly invoke the results from that work. In particular, as $\mathcal{X}$ is assumed

to be convex, the intrinsic metric used in (Qu et al., 2023) reduces to the Euclidean metric. From $KV \leq V - U$, we have

$$KV \leq V - U \leq V - U \cdot V/\sup V \leq rV, \quad r = 1 - \inf U/\sup V.$$

By Theorem 3 in (Qu et al., 2023),

$$W(X_n, X_\infty) \leq \frac{1}{\inf V} \frac{r^n}{1-r} \mathbb{E} \|X_0 - X_1\| V(X_0 + \tilde{U}(X_1 - X_0)) = Cr^n$$

where

$$C = \frac{\mathbb{E} \|X_0 - X_1\| V(X_0 + \tilde{U}(X_1 - X_0))}{\inf U \cdot (\inf V/\sup V)}$$

and $\tilde{U}$ is a $U[0,1]$ random variable independent of $X_0$ and $X_1$. $\qquad\square$

*Proof of Theorem 4.* The proof is similar to the proof of Theorem 1 in (Qu et al., 2023), but in our specific setting, the proof becomes much simpler notation-wise. As in the proof of our Theorem 1, we have

$$V_1(x) \geq \sum_{n=0}^{\infty} K^n V_0(x), \quad K^n V_0(x) = \mathbb{E}_x \left[ V_0(X_n) \prod_{l=1}^{n} Df_l(X_{l-1}) \right].$$

Following the same induction process as in (Qu et al., 2023), we have

$$V_m(x) \geq \sum_{n=0}^{\infty} c_{m,n} K^n V_0(x), \quad c_{m,n} = \binom{n+m-1}{m-1}.$$

Let $F_n = f_n \circ \cdots \circ f_1$ and $\bar{F}_n = f_1 \circ \cdots \circ f_n$ be the $n$-fold forward and backward composition, respectively. Given $f$ independent of anything else,

$$\begin{aligned}
\int_x^{f(x)} V_m(y)dy &\geq \sum_{n=0}^{\infty} c_{m,n} \int_x^{f(x)} K^n V_0(y)dy \\
&= \sum_{n=0}^{\infty} c_{m,n} \mathbb{E} \int_x^{f(x)} \left[ V_0(F_n(y)) \prod_{l=1}^{n} Df_l(F_{l-1}(y)) \right] dy \\
&\geq \sum_{n=0}^{\infty} c_{m,n} \mathbb{E} \int_{F_n(x)}^{F_n(f(x))} V_0(z)dz \\
&\geq \sum_{n=0}^{\infty} c_{m,n} \mathbb{E} \int_{\bar{F}_n(x)}^{\bar{F}_{n+1}(x)} V_0(z)dz
\end{aligned}$$

For a particular $\bar{n}$,

$$\begin{aligned}
\int_x^{f(x)} V_m(y)dy &\geq \sum_{n=\bar{n}}^{\infty} c_{m,n} \mathbb{E} \int_{\bar{F}_n(x)}^{\bar{F}_{n+1}(x)} V_0(z)dz \\
&\geq c_{m,\bar{n}} \mathbb{E} \int_{\bar{F}_{\bar{n}}(x)}^{\bar{F}_\infty(x)} V_0(z)dz \\
&\geq c_{m,\bar{n}} \inf V_0 \cdot \mathbb{E} \left\| \bar{F}_n(x) - \bar{F}_\infty(x) \right\|.
\end{aligned}$$

By integrating with respect to the initial distribution $X_0$,

$$\begin{aligned}
\mathbb{E} \|X_0 - X_1\| V_m(X_0 + \tilde{U}(X_1 - X_0)) &\geq c_{m,\bar{n}} \inf V_0 \cdot \mathbb{E} \left\| \bar{F}_{\bar{n}}(X_0) - \bar{F}_\infty(X_0) \right\| \\
&\geq c_{m,\bar{n}} \inf V_0 \cdot W(X_{\bar{n}}, X_\infty) \\
&= \frac{(\bar{n}+m-1)\dots(\bar{n}+1)}{(m-1)\dots 1} \inf V_0 \cdot W(X_{\bar{n}}, X_\infty) \\
&= \prod_{k=1}^{m-1} \left( \frac{\bar{n}}{k} + 1 \right) \inf V_0 \cdot W(X_{\bar{n}}, X_\infty).
\end{aligned}$$

$\qquad\square$

*Proof of Theorem 5.* As $\mathcal{X}$ is compact, without loss of generality, let $\mathcal{X} = [0,1]^d$. The main goal is to find $M, N$ such that

$$P\left(\sup_{x\in\mathcal{X}} [KV(x) - V(x) + U(x)] > \sup_{x\in\mathcal{M}} \left[\hat{K}_N V(x) - V(x) + U(x)\right] + \epsilon\right) \leq \delta.$$

This probability is bounded by the sum of

$$P\left(\sup_{x\in\mathcal{X}} [KV(x) - V(x) + U(x)] > \sup_{x\in\mathcal{M}} [KV(x) - V(x) + U(x)] + \epsilon/2\right) \tag{5}$$

and

$$P\left(\sup_{x\in\mathcal{M}} [KV(x) - V(x) + U(x)] > \sup_{x\in\mathcal{M}} \left[\hat{K}_N V(x) - V(x) + U(x)\right] + \epsilon/2\right). \tag{6}$$

To bound (5), we need to bound the Lipschitz constant of $KV - V + U$. For $x, y \in \mathcal{X}$,

$$\begin{aligned}
&|KV(x) - KV(y)| \\
&= |\mathbb{E}Df(x)V(f(x)) - \mathbb{E}Df(y)V(f(y))| \\
&\leq |\mathbb{E}Df(x)V(f(x)) - \mathbb{E}Df(x)V(f(y))| + |\mathbb{E}Df(x)V(f(y)) - \mathbb{E}Df(y)V(f(y))| \\
&\leq \mathbb{E}Df(x)\,|V(f(x)) - V(f(y))| + \mathbb{E}\,|Df(x) - Df(y)|\,V(f(y)) \\
&\leq DV \cdot \mathbb{E}Df^2 \cdot \|x - y\| + \sup V \cdot \mathbb{E}D^2 f \cdot \|x - y\|
\end{aligned}$$

where $DV$ is the Lipschitz constant of $V$. Then, the Lipschitz constant of $KV - V + U$ is bounded by

$$\tilde{L} \triangleq DV \cdot \mathbb{E}Df^2 + \sup V \cdot \mathbb{E}D^2 f + DV + DU.$$

Then (5) is bounded by $P\left([0,1]^d \not\subset \cup_{x\in\mathcal{M}} B_{\tilde{r}}\right)$ where $\tilde{r} = \epsilon/(2\tilde{L})$. To bound this probability, we divide the unit cube into $\tilde{C} = (\sqrt{d}/\tilde{r})^d = (2\tilde{L}\sqrt{d}/\epsilon)^d$ sub-cubes with edge length $\tilde{r}/\sqrt{d}$. Then $[0,1]^d \not\subset \cup_{x\in\mathcal{M}} B_{\tilde{r}}$ implies that there exists at least one sub-cube that does not contain any element of $\mathcal{M}$. This is equivalent to failing to collect $\tilde{C}$ different coupons within $M$ draws. It is well-known that we need $\Theta(\tilde{C}\log\tilde{C})$ on average to collect $\tilde{C}$ different coupons. By Markov inequality, we can choose $M = O(\tilde{C}\log\tilde{C}/\delta) = O(\log(1/\epsilon)/(\delta\epsilon^d))$ to reduce the failure probability below $\delta/2$.

To bound (6), let $x_\mathcal{M} = \arg\max_{x\in\mathcal{M}} [KV(x) - V(x) + U(x)]$. By Chebyshev inequality,

$$\begin{aligned}
&P\left(KV(x_\mathcal{M}) - V(x_\mathcal{M}) + U(x_\mathcal{M}) > \sup_{x\in\mathcal{M}} \left[\hat{K}_N V(x) - V(x) + U(x)\right] + \epsilon/2 \Big| \mathcal{M}\right) \\
&\leq P\left(KV(x_\mathcal{M}) - V(x_\mathcal{M}) + U(x_\mathcal{M}) > \hat{K}_N V(x_\mathcal{M}) - V(x_\mathcal{M}) + U(x_\mathcal{M}) + \epsilon/2 \Big| \mathcal{M}\right) \\
&= P\left(KV(x_\mathcal{M}) > \hat{K}_N V(x_\mathcal{M}) + \epsilon/2 \Big| \mathcal{M}\right) \\
&\leq \frac{\text{Var}\left[Df(x_\mathcal{M})V(f(x_\mathcal{M}))|\mathcal{M}\right]/N}{\epsilon^2/4} \\
&\leq \frac{4\sup V^2 \cdot \mathbb{E}Df^2}{N\epsilon^2} \\
&\leq \delta/2
\end{aligned}$$

when $N \geq (8\sup V^2 \cdot \mathbb{E}Df^2)/(\delta\epsilon^2) = O(1/(\delta\epsilon^2))$.

$\square$

*Proof of Proposition 1.* By $X_0 = 0$ and (3), we have

$$X_n = \sum_{k=1}^n H^{n-k} Z_k \quad \text{and} \quad \bar{X}_n = \sum_{k=1}^n H^{k-1} Z_k$$

where $\bar{X}_n$ is the backward chain. By definition,

$$W(X_n, X_\infty) \leq \mathbb{E}\left\|\bar{X}_\infty - \bar{X}_n\right\| = \mathbb{E}\left\|\sum_{k=n+1}^\infty H^{k-1} Z_k\right\| = \mathbb{E}\left\|H^n \sum_{k=1}^\infty H^{k-1} Z_k\right\| = \mathbb{E}\left\|H^n Y\right\|.$$

Let Lip(1) be the family of functions on $\mathbb{R}^d$ that are 1-Lipschitz with respect to the 2-norm $\|\cdot\|$. Let $\|\cdot\|_1$ be the 1-norm. By the Kantorovich–Rubinstein duality,

$$
\begin{aligned}
W(X_n, X_\infty) &= \sup_{h \in \text{Lip}(1)} |\mathbb{E}h(X_\infty) - \mathbb{E}h(X_n)| \\
&\geq \mathbb{E}\left[\left\|\bar{X}_\infty\right\|_1 - \left\|\bar{X}_n\right\|_1\right]/\sqrt{d} \\
&= \mathbb{E}\left\|\bar{X}_\infty - \bar{X}_n\right\|_1/\sqrt{d} \\
&\geq \mathbb{E}\left\|\bar{X}_\infty - \bar{X}_n\right\|/\sqrt{d} \\
&= \mathbb{E}\left\|H^n Y\right\|/\sqrt{d},
\end{aligned}
$$

where the second line is because $\|\cdot\|_1/\sqrt{d}$ is 1-Lipschitz with respect to $\|\cdot\|$, the third line is because $\bar{X}_\infty \geq \bar{X}_n \geq 0$, and the fourth line is because $\|\cdot\|_1 \geq \|\cdot\|$. Let $\Lambda$ be the diagonal matrix containing all the eigenvalues of $H$. Let $Q$ be the orthogonal matrix containing all the (column) eigenvectors of $H$. Then

$$
H^n = Q\Lambda^n Q^\top = \sum_{i=1}^{d} \lambda_i^n q_i q_i^\top.
$$

Recall that $\lambda$ is the largest eigenvalue of $H$. Then

$$
\mathbb{E}\left\|H^n Y\right\|/\lambda^n = \mathbb{E}\left\|\sum_{i=1}^{d}(\lambda_i/\lambda)^n q_i q_i^\top Y\right\| \to \mathbb{E}\left\|\sum_{i \in I} q_i q_i^\top Y\right\|, \quad \text{as } n \to \infty
$$

where $\{q_i : i \in I\}$ are the eigenvectors corresponding to $\lambda$. Since $Z$ is integrable and $Y$ does not concentrate on the orthogonal complement of the eigenspace associated with $\lambda$, the above limit is finite and positive.

$\square$

