# OpenReview forum: "Deep Learning for Computing Convergence Rates of Markov Chains"
_NeurIPS.cc/2024/Conference — NeurIPS 2024 spotlight_

### Official Review · Reviewer_3Pb4 · 2024-07-08

**Soundness:** 3
**Presentation:** 2
**Contribution:** 3
**Rating:** 6
**Confidence:** 3

**Summary:**

The authors proposed a novel computational method to estimate the convergence rate of general Markov Chains. They utilized neural network to verify if contration drift (CD) holds for a given Markov Chain. As an extension to a prior work [1], the authors provided further theoretical analysis of their methods, and proposed an explicit formula of the convergence rate. Various numerical experiments were conducted to justify the applicability.

**Strengths:**

1. I'm not familiar with the topic about estimating the convergence rate of a given Markov chain. In general, the idea is very novel to me. Since this problem is very difficult, it seems that the proposed method is indeed promising.

2. The paper provides detailed theoretical analysis, including the explicit formula of convergence rate and sample complexity.

**Weaknesses:**

1. The paper is somewhat intricate and difficult follow. I notice that it is built upon [1], but more detailed background and related work should be provided to improve readability. Specifically, the authors should discuss more previous methods if there exist and make comparisons theoretically or numemrically.

2. In experiments, the authors only showed the results of their algorithms. I think the convergence rate should be verifed through directly simulating the Markov chain and computing Wasserstein distance between the sampled distribution and stationary distribution.

3. The authors only conducted experiments in 2D, lacking high dimensional examples.

4. A more fundamental problem is that how to verify whether CD holds in practice in high dimensional case. The inequality is defined in the pointwise sense, which is difficult to verify due to the curse of dimensionality.

[1] Qu, Yanlin, Jose Blanchet, and Peter Glynn. "Computable Bounds on Convergence of Markov Chains in Wasserstein Distance." arXiv preprint arXiv:2308.10341 (2023).

**Questions:**

1. What is the relationship between CD and conventional inequailities such as Poincare inequality, Log Sobolev inequality?

1. I wonder the choices of $U$. In all the experiments, $U$ is fixed as a constant. How is this constant selected? Also, is there any other choice of $U$ that can be considered in practice? How does different choices of $U$ lead to different convergence rate estimation? What is the practicability of sequentially learning neural network in section 3.4?

2. The convergence rate in Theorem 3 depends on $\inf/\sup V$. How is inf/sup $V_\theta$ computed in practice?

3. Does there always exist $U,V$ to satisfy CD? If not, how can one know that from DCDC? In another word, how can one tell the failure of training of network from the fact that CD doesn't hold?

**Limitations:**

The limitations are pointed out in the weakness and question part.

---

> ### Author Rebuttal · Authors · 2024-08-07
>
> We sincerely thank you for your detailed feedback. In the following, we address the concerns (W1-4) and answer the questions (Q1-4).
>
> W1. In Section 2, we only introduce necessary *analytical* concepts (e.g., random mapping representation, local Lipschitz constant, and contractive drift) to quickly set up for building the *computational* framework. In retrospect, we agree that we should add more background (e.g., Markov chains, stationary distributions, and traditional convergence analysis) to enhance the readability. This will be done in the revision.
>
> Since DCDC is the first computational framework to bound the convergence of general state-space Markov chains, there are currently no other numerical methods available for direct comparison. Since analytical methods can only handle *stylized* (structured) Markov chains, the *realistic* (less structured) examples considered in this paper are clearly beyond their reach.
>
> W2. Given a non-trivial Markov chain $X$, simulating $X_n$ for large $n$ is often very expensive, simulating $X_\infty$ directly is typically not feasible, and the convergence rate ($r<1$ with $C>0$ such that $W(X_n,X_\infty)\leq Cr^n$) is determined by the *infinite* sequence $W(X_0,X_\infty)$, $W(X_1,X_\infty)$, …. These three reasons make it impossible to reliably estimate the convergence rate via direct simulation in finite time.
> Now we have DCDC to generate a convergence bound $W(X_n,X_\infty)\leq Cr^n$, the correctness of which is theoretically guaranteed, so it is not necessary to verify the bound by estimating $W(X_n,X_\infty)$. Although estimating $W(X_n,X_\infty)$ can reveal whether the above bound is tight, it is often impractical due to the first two reasons mentioned earlier. In particular, for the examples in this paper, $X_\infty$ is unknown. In fact, complex dynamics + intractable equilibrium = notoriously hard convergence analysis (without DCDC).
>
> W3. We plan to apply DCDC to high-dimensional chains in future work. In the current paper, we focus on 2D examples to verify the effectiveness, visualize the CDE solution, and gain valuable insights from the shape (e.g., a sunken surface, a sloping plane, and a wedge-like curve).
>
> W4. The pointwise verification of an equality/inequality in a high dimensional space is indeed a fundamental issue. These issues, however, are common to other areas for which NN solvers of functional equations have demonstrated huge success [5]. For instance, Neural-network-based PDE solvers also face this same issue. While the sample complexity literature typically uses an L2-based criterion (based on suitable Sobolev norms), see, e.g., [6], typical applications often require approximations of PDE solution with uniform convergence guarantees on a specific set of points. While we expect dimension-dependent complexity rates as in the PDE literature, we note that our task is easier because we only need to prove an inequality, not an equality. However, given the successful record and vast literature of PINNs, we believe that our method can be at least equally successful in a wide range of settings and we plan to pursue a sharp sample complexity theory similar to that developed in the PDE literature in future research.
>
> Q1. There are two primary classes of methods to bound the convergence of Markov chains: drift & minorization/contraction conditions (Chapter 9-20 of [2]) and spectral/operator theory (Chapter 22 of [2]). Poincare and log-sobolev inequalities belong to the latter while CD in [1] advances the former. “The former have been the most successful for the study of stability and convergence rates, despite the inherent difficulty of constructing an appropriate Lyapunov function” [3]. DCDC leverages deep learning to tackle this inherent difficulty (and more).
>
> Q2. When $U$ is a constant, the constant is theoretically not important as CD is linear in $V$. However, in practice, we can use this constant to control the scale of $V$. In the SGD example, $KV=V-1$ leads to $\max V\approx1000$, so we use $0.1$ to scale $V$ down, which turns out to be easier to learn/approximate.
>
> When $U$ is not a constant, it modifies the underlying metric (Section 2 of [1]). When a chain is expansive ($d(f(x),f(y))>d(x,y)$), we may find $U$ to make it non-expansive ($d_U(f(x),f(y))\leq d_U(x,y)$). The examples in this paper are already non-expansive, so we set $U$ to be constant. The application of DCDC to expansive chains is left for future research where sequential CDE solving (Section 3.4) becomes crucial.
>
> Q3. In this paper, the infimum and supremum are computed over a mesh grid. The error can be controlled if the Lipschitz constant of the neural network is estimated (e.g., [4]).
>
> Q4. Given $U$, $V$ is an expected discounted cumulative reward (Remark in Section 2), so it always exists but can be infinite. When $V=\infty$, the neural network in DCDC diverges to infinity. When it does not diverge to infinity, the training is successful if we verify the CDE.
>
>
> [1] Qu, Y., Blanchet, J., Glynn, P., “Computable bounds on Convergence of Markov chains in Wasserstein distance”, 2023
>
> [2] Andrieu, C., Lee, A., Power, S., Wang, A.Q., “Comparison of Markov chains via weak Poincare inequalities with application to pseudo-marginal MCMC”, 2022
>
> [3] Douc, R., Moulines, E., Priouret, P., Soulier, P., “Markov Chains”, 2018
>
> [4] Scaman, K., Virmaux, A., “Lipschitz regularity of deep neural networks: analysis and efficient estimation”, 2018
>
> [5] Raissi, M., Perdikaris, P., Karniadakis, G.E., “Physics-informed neural networks: A deep learning framework for solving forward and inverse problems involving nonlinear partial differential equations”, 2018
>
> [6] Lu, Y., Blanchet, J., Ying, L., “Sobolev Acceleration and Statistical Optimality for Learning Elliptic Equations via Gradient Descent”, 2021

---

> > ### Comment · Reviewer_3Pb4 · 2024-08-08
> >
> > Thanks for you feedback addressing most of my concerns. I still believe some more experiments mentioned in W2 can be conducted to strengthen the paper. I understand the expensive cost of simulation a general non-trivial Markov chain, but you can try some easier one. For instance, in the SGD for logistic regression case, the computational cost is low and thus one can run sufficienly many steps (e.g. 100k) to get sufficiently accurate estimation of $X_\infty$. Then the true convergence rate can be computed (at least approximated) and compared with the theoretical results. Nevertheless, I think this work is indeed interesting and constructive after reading the author rebuttal. I'm happy to raise my score.

---

> > > ### Author Response · Authors · 2024-08-08
> > >
> > > Thank you for your reply. The suggestion of trying easier Markov chains for exact rates is very helpful and we will include this comparison. Since we mainly care about whether our bounds are good for reasonably large $n$’s (but not the first several $n$’s), given the exponential convergence, we may need to estimate some very small $W(X_n,X_\infty)$, which is pretty hard in general, but we could do it efficiently in a non-trivial multidimensional process, using importance sampling.

---

### Official Review · Reviewer_wP6n · 2024-07-08

**Soundness:** 4
**Presentation:** 4
**Contribution:** 4
**Rating:** 8
**Confidence:** 3

**Summary:**

The paper studies the problem of convergence rate analysis for general state-space Markov chains. They propose Deep Contractive Drift Calculator (DCDC), the first general-purpose sample-based algorithm for bounding the convergence of Markov chains. There are two components, a theoretical one that utilize an auxiliary function (a solution of a certain equation) to bound convergence, and an empirical one that utilize deep neural networks to approximate the auxiliary function. Furthermore, the authors provide statistical guarantees on the approximation.

**Strengths:**

1. Extremely well written paper. The abstract gives a quite clear overview of the paper, pointing out the main contribution in this paper and the importance of the problem addressed without exaggeration. The wording in the paper is succinct, subjective, yet pleasant to read. I'm not an expert in the field but I quickly understand the importance of the paper. It appears to me that the whole presentation is rather mature and professional.
2. The authors study a problem of fundamental importance and give a general solution that is simple yet effective and without sound guarantees. The authors borrow the idea of Lyapunov function approximated by neural networks and successfully apply it to the convergence problem of Markov chains, which in my opinions is of some fundamental importance. Furthermore, the authors provide statistical guarantees on the approximation, forming a rather complete story.

**Weaknesses:**

I'm not able to find effective weaknesses in this paper.

**Questions:**

No.

**Limitations:**

I'm not seeing any limitations of essence not addressed in this paper.

---

> ### Author Rebuttal · Authors · 2024-08-07
>
> We sincerely thank you for your positive feedback. Please feel free to read the other rebuttals.

---

> > ### Comment · Reviewer_wP6n · 2024-08-07
> > **Thank you**
> >
> > Thank you for your positive feedback on my positive feedback. I'll read the other rebuttals.

---

### Official Review · Reviewer_ituE · 2024-07-13

**Soundness:** 3
**Presentation:** 3
**Contribution:** 3
**Rating:** 8
**Confidence:** 3

**Summary:**

#### Summary
The paper introduces the Deep Contractive Drift Calculator (DCDC), a novel sample-based algorithm for bounding the convergence rates of general state-space Markov chains to stationarity in Wasserstein distance. The method leverages deep learning to solve the Contractive Drift Equation (CDE), providing explicit convergence bounds. The paper includes theoretical analysis, sample complexity, and empirical validation on realistic Markov chains.

**Strengths:**

#### Strengths
1. **Innovative Approach**: The use of deep learning to solve the Contractive Drift Equation (CDE) is novel and bridges a gap between deep learning and traditional mathematical analysis.
2. **Theoretical Rigor**: The paper provides thorough theoretical foundations, including the derivation of the CDE and detailed proofs of convergence bounds.
3. **Practical Implications**: The approach is validated on realistic Markov chains, demonstrating its applicability to problems in operations research and machine learning.
4. **Clarity of Exposition**: The paper is well-written, with clear explanations of the methodology and theoretical results.

**Weaknesses:**

#### Weaknesses
1. **Computational Complexity**: The approach may be computationally intensive, particularly for high-dimensional state spaces. More discussion on computational efficiency and scalability would be beneficial.
2. **Comparison with Existing Methods**: While the paper discusses theoretical advantages, empirical comparisons with existing state-of-the-art methods for convergence analysis are limited.
3. **Generality**: The method is demonstrated on specific types of Markov chains. Extending the empirical validation to a broader range of applications would strengthen the paper.
4. **Sample Complexity**: Although the sample complexity is analyzed, practical guidelines for choosing sample sizes in different scenarios would enhance the utility of the method.

**Questions:**

#### Questions
1. **Computational Complexity**:
    - Could you provide more details on the computational complexity of the DCDC algorithm, particularly for high-dimensional state spaces? How does the method scale with increasing dimensions?

2. **Comparison with Existing Methods**:
    - How does the DCDC method compare empirically with existing state-of-the-art methods for bounding convergence rates of Markov chains? Are there specific scenarios where DCDC significantly outperforms these methods?

3. **Generality**:
    - The method is validated on specific types of Markov chains. Do you foresee any challenges in applying DCDC to other types of Markov chains, such as those with more complex dynamics or in higher-dimensional spaces?

4. **Sample Complexity**:
    - While you provide a theoretical analysis of sample complexity, can you offer practical guidelines or heuristics for choosing sample sizes in different applications? How sensitive is the method to the choice of sample size?

5. **Practical Applications**:
    - Can you discuss potential practical applications of the DCDC method in more detail? For instance, how might this method be applied in real-world scenarios such as reinforcement learning or stochastic optimization?

6. **Assumptions and Limitations**:
    - The paper discusses some assumptions and limitations. Could you elaborate on the key assumptions that are critical for the theoretical results, and how robust the method is to violations of these assumptions?

---

> ### Author Rebuttal · Authors · 2024-08-07
>
> We thank you for your valuable feedback and positive view about our paper. In the following, we address the concerns (1-4) and answer the questions (1-6).
>
> 1&4 *Computational and Sample Complexity*: The computational complexity, as with any typical deep learning method involving general non-convex optimization, is largely an open problem. There are, however, recent results for regularized ReLU architectures which can be studied to global convergence using convex relaxations (see [1]). We are interested in exploring the applications of these results in future work. We believe that we can use the parameter $\epsilon$ to control the complexity of these convex relaxations relative to the inequality-gap in the CD bound induced in terms of the parameter $\epsilon$. It is important to keep in mind (as we mention in the paper) that our task is simpler than finding an exact solution, because we care about a one-sided inequality. In terms of sample complexity, the curse of dimensionality is also, unfortunately, an issue that plagues virtually any algorithm that learns a function by sampling [2]. Our goal in this paper is to introduce a novel methodology (the first of its type) that enables the use of deep learning to address this important problem, but we admit that obtaining sharp sample complexity bounds is an important problem that we are also leaving for future research. We envision a sample complexity theory that depends on $\epsilon$ in such a way that as $\epsilon$ is small we recover the sample complexity guarantees that are expected if we know that the CDE is similar to the analysis in [3]. We also leave this important topic for future research.
>
> 2 *Comparison with Existing Methods*:
> We are not aware of other data driven computational frameworks to bound the convergence of general state-space Markov chains. Since analytical methods can only handle *stylized* (structured) Markov chains, the *realistic* (less structured) examples considered in this paper are clearly beyond the reach of the existing methods (which are based on analytical developments that are not “automatic” or computer based). We can provide a discussion, however, between CD and other existing methods to build the inequalities per-se, this discussion will summarize the comparison presented in [4].
>
> 3 *Generality*:
> Markov chains find important applications in a wide range of disciplines (including Computer Science, Economics, Electrical Engineering, Management Science, Operations Research etc.). We use non-trivial examples in Operations Research (e.g., queueing networks) and Machine Learning (e.g., stochastic gradient descent) to illustrate the applicability of the method. For complex Markov chains (e.g., reflected Brownian motions) in high dimensional spaces, one challenge is that contraction may occur along some but not all directions (e.g., $|\partial f/\partial x_1|<1$ or $|\partial f/\partial x_2|<1$ but not both), resulting in a local Lipschitz constant of one. In our ongoing work, we will introduce vector-valued CDs to address this issue.
>
> 5 *Practical Applications*:
> Thanks for this important question. Regarding the application to stochastic optimization, while we include an SGD example, admittedly this is just to show-case the applicability of the method. While the example that we provide already saturates what can be done with “standard methods” (which again, virtually all involve pen-and-paper approaches) we recognize that a broader ablation is needed to fully understand the potential of this approach. For the application to reinforcement learning, note that typical results mostly focus on finite state spaces using assumptions that are rather difficult to verify in general state spaces. Our methods open up the development of algorithms that satisfy CD type conditions. We will also mention this in the camera ready version
>
> 6 *Assumptions and Limitations*:
> The current paper focuses on compact spaces. The extension to non-compact spaces is left for future research. On compact spaces, the key assumption is CD itself (i.e., CD has a solution). As discussed at the end of Section 2, the CDE solution is an expected discounted cumulative reward, so it exists but can be infinite. When it is infinite, the chain has too little contraction to converge in Wasserstein distance. In this case, the neural network in DCDC diverges to infinity, which can be viewed as a certificate of non-convergence.
>
> [1] Ergen, T., Pilanci, M., “Global optimality beyond two layers: Training deep ReLU networks via convex programs”, 2021
>
> [2] Raissi, M., Perdikaris, P., Karniadakis, G.E., “Physics-informed neural networks: A deep learning framework for solving forward and inverse problems involving nonlinear partial differential equations”, 2018
>
> [3] Lu, Y., Blanchet, J., Ying, L., “Sobolev Acceleration and Statistical Optimality for Learning Elliptic Equations via Gradient Descent”, 2021
>
> [4] Qu, Y., Blanchet, J., Glynn, P., “Computable bounds on Convergence of Markov chains in Wasserstein distance”, 2023

---

### Author Rebuttal · Authors · 2024-08-07

* We appreciate the feedback and comments of all of the referees. We note that two out of the three reports rate the paper with an evaluation of 8 (strong accept) whereas one of the referees has some concerns providing an evaluation of 4 (borderline reject).
* We try to focus most of the response below to answer questions raised and to address concerns raised. The main issues have to do with the existence of a solution to the CD, the answer is yes, but it may diverge and this could be practically detected in training. The second issue has to do with complexity results and the fact that learning a function in high dimensions is subject to the curse of dimensionality. But this issue is present in every single application of deep learning, which in the end involves approximating high dimensional functions based on a limited sample.
* The bottom line is that this paper is the first one that enables the use of deep learning to estimate rates of convergence to stationarity for Markov chains that take values on a general state space. We acknowledge the limitations and we’ll be happy to add more discussion (along the lines of what we include in this report, taking as a template the literature on solutions to PDEs based on deep learning techniques).

---

### Comment · Area_Chair_gW3B · 2024-08-08
**Please read the authors’ rebuttal and reply by August 13, 2024 (11:59 PM, AOE time)**

Dear Reviewers,

Thank you for your hard work during the review process. The authors have responded to your initial reviews. **If you haven’t already done so, please take the time to carefully read their responses.** It is crucial for the authors to receive acknowledgment of your review by the deadline for the author-reviewer discussion period, which is August 13, 2024 (11:59 PM, Anywhere on Earth). Please address any points of disagreement with the authors as early as possible.

Best,

Your AC

---

### Decision · Program_Chairs · 2024-09-25

**Decision:**

Accept (spotlight)

**Comment:**

This paper proposes a general-purpose deep learning-based algorithm to compute the convergence rate of Markov chains to their stationary distribution. The proposed method consists of two components:

1. **Theoretical Tool:**
   The first component is a theoretical tool that demonstrates how the convergence rate of the Markov chain can be exactly determined by solving a Contractive Drift Equation (CDE), derived from the contractive drift condition of the Markov chain.

2. **Network-Based Solver:**
   The second component is a network-based solver for the CDE equation. The authors provide examples on Stochastic Gradient Descent (SGD) and other practical Markov chains to showcase how the convergence rate can be obtained for complex, real-world chains.

All reviewers agree that this work is well-written and presents a novel solution that bridges the gap between the theoretical analysis of Markov chains and data-driven algorithms. As suggested by the reviewers, it would be beneficial to include refined experiments that verify the convergence rate by running sufficient steps of the Markov chains.

Overall, this paper is a clear acceptance, and the idea of using deep learning to estimate the convergence of Markov chains deserves more attention from the community.